# COVID-19 Global Risk: Expectation vs. Reality

**DOI:** 10.3390/ijerph17155592

**Published:** 2020-08-03

**Authors:** Mudassar Arsalan, Omar Mubin, Fady Alnajjar, Belal Alsinglawi

**Affiliations:** 1School of Computer, Data and Mathematical Sciences, Western Sydney University, Sydney 2116, Australia; mharsalan@gmail.com (M.A.); o.mubin@westernsydney.edu.au (O.M.); B.Alsinglawi@westernsydney.edu.au (B.A.); 2College of Information Technology, UAE University, Al-Ain, UAE

**Keywords:** COVID-19, risk evaluation, multi-weighted factor analysis

## Abstract

*Background and Objective*: COVID-19 has engulfed the entire world, with many countries struggling to contain the pandemic. In order to understand how each country is impacted by the virus compared with what would have been expected prior to the pandemic and the mortality risk on a global scale, a multi-factor weighted spatial analysis is presented. *Method*: A number of key developmental indicators across three main categories of demographics, economy, and health infrastructure were used, supplemented with a range of dynamic indicators associated with COVID-19 as independent variables. Using normalised COVID-19 mortality on 13 May 2020 as a dependent variable, a linear regression (N = 153 countries) was performed to assess the predictive power of the various indicators. *Results*: The results of the assessment show that when in combination, dynamic and static indicators have higher predictive power to explain risk variation in COVID-19 mortality compared with static indicators alone. Furthermore, as of 13 May 2020 most countries were at a similar or lower risk level than what would have been expected pre-COVID, with only 44/153 countries experiencing a more than 20% increase in mortality risk. The ratio of elderly emerges as a strong predictor but it would be worthwhile to consider it in light of the family makeup of individual countries. *Conclusion*: In conclusion, future avenues of data acquisition related to COVID-19 are suggested. The paper concludes by discussing the ability of various factors to explain COVID-19 mortality risk. The ratio of elderly in combination with the dynamic variables associated with COVID-19 emerge as more significant risk predictors in comparison to socio-economic and demographic indicators.

## 1. Introduction

The Coronaviridae (HCoV) family are historically known to be responsible for a large proportion of common colds and other upper respiratory tract infections. HCoV are now known to be associated with several serious respiratory diseases, i.e., bronchitis, bronchiolitis, or pneumonia, especially affecting elderly people and immunosuppressed patients. They have also been prevalent in nosocomial viral infections [1]. At the end of December 2019, a novel coronavirus disease (COVID-19) was reported from China [2]. The virus causing COVID-19, named severe acute respiratory syndrome coronavirus-2 (SARS-CoV-2), quickly spread across the globe. As of early July 2020, more than 12 million people have been infected with more than 500,000 deaths [3]. The burden of the disease is increasing and impacting the global economy severely [4]. The absence of any vaccine to control COVID-19 has introduced numerous challenges for governments and health organisations [5]. Hence, the impact of such a pandemic has severely affected not only the economy but also the health systems of a large number of countries. The demographic attributes of any country such as population size and density, age structure, gender balance, etc., and socio-economic factors such as income, expenditure, and education can ultimately dictate not only how a virus may spread in a pandemic but also how effective each country’s response is against the virus [6]. The prevailing infrastructure in a country will influence how well it is able to mitigate the impact of the virus and save lives. The development of a country’s response or its required infrastructure is dependent upon the adoption of an overview or holistic “risk” analysis using each contributing factor, thereby enabling relevant agencies in those countries to pre-empt interventions or upgrades in specific areas and be better prepared to manage the pandemic. Therefore, the primary research aim of the analysis described in this paper was to determine or assess the level of mortality risk (both current as well as expected) of COVID-19 on a global scale in evolving circumstances using a multi-factor weighted strategy, where risk is defined as the probability and consequences of an unwanted and negative sequence of events (such as the COVID-19 pandemic) exacted upon a country [7].

In prior literature, a number of interesting case studies have been presented that analyse the numerous demographic and socio-economic risk factors that influence the epidemiology of various pandemics or epidemics other than COVID-19. For example, population density is considered as a risk evaluation or “forecasting” measure in infectious diseases, driving economic growth and spending, impacting spatial spread [6] as well as dictating how practical social distancing measures can be in various regions [8]. The various strains of the influenza virus have been thoroughly studied in light of these and other socio-economic factors. Income, race, education, and spatial characteristics (such as where patients live; specific country, region, or suburb) have been shown to significantly be associated with the death rate of influenza (Type A) [9,10].Drops in temperature and rainfall have shown to correlate with the rate of increase in influenza (Type A) cases [11]. An independent study undertaken on the spread of pandemic influenza on board a New Zealand troop ship indicated that the age group of 25–34-year-old sailors displayed an increased mortality risk compared with those under 25 years of age [12]. Standard of living is also an important risk factor linked with the spread of an epidemic, as seen in the case of tuberculosis [13] or HIV [14]. Similar outcomes were observed with Ebola [15] where a comprehensive set of indicators, including social, economic, and cultural factors were proposed to create a risk evaluation and assessment framework for the virus. Such socio-economic risk factors are also understood to play a significant role in the emergence of other non-contagious conditions such as obesity [16]. With the rapid spread of COVID-19, different countries have adopted different response strategies to contain the spread and minimise the death toll. The efficacy of these interventions and the impact of COVID-19 is strongly associated with a number of socio-economic factors such as the vulnerability of the population [17]. Therefore, the aim of the work presented in this paper is to assess the mortality risk level of each country to COVID-19 as determined by a collage of static socio-economic factors, in combination with a range of dynamic epidemiological indicators.

Prior work has attempted to understand the correlation between certain risk elements such as demographic variables and certain specific socio-economic factors (such as GDP and population density) towards COVID-19. Whilst this work was only restricted to specific regions or continents such as China [18] or Europe [19], ultimately a global overview can be very insightful and telling. In the latter study for example, the predictive power of 10 socio-economic and demographic factors was analysed at different times in March 2020 for different European countries. Although this temporal layer is very welcome, a pre-COVID baseline risk determination is absconding. A similar study was conducted for Africa [20]. Therefore, as far as it is understood, a global overview or mortality risk evaluation of COVID-19 is absent with respect to a diverse range of socio-economic factors coupled with dynamic factors. Consequently, the aim of the analysis presented in this manuscript was to model a change in risk to establish whether countries were currently at a level of mortality risk consistent with their expectations.

An analysis of prior work in the area of multi-criteria and factor association on pandemics revealed that demographics, economy, and health attributes were amongst the most commonly-used predictors of risk. Hence, three primary categories of factors were selected, namely: Demography (including total population, ratio of elderly above 65, population density, and disability-adjusted life years), economy (including gross domestic product, health expenditure), and health infrastructure (including total hospital beds, total physicians, and total nurses). This provided a total of nine static risk factors. These static indicators were used to compute a base mortality risk on a global scale. Further in the analysis, these risk factors were complimented with an additional six dynamic factors related to current COVID-19 spread (as at 13 May 2020), including current active cases, current susceptible population, three growth rates (recovered, confirmed, and mortality), and the prevailing stringency level imposed by the government in the country of interest.

## 2. Method

In this section, data collection and modelling aspects around the multi-factor weighted spatial analysis are described. The methodology of this paper is grounded in prior work, where regression and multi-factor analysis were used to assess the importance and risks of numerous variables in an epidemic [21].

### 2.1. Data Collection

Data was collected from multiple international organisations including the World Bank (WB), National Aeronautics and Space Administration (NASA), Johns Hopkins University and Medicine (JHU), and Environmental Systems Research Institute (ESRI). The dataset of the latest world political boundaries was downloaded from ESRI and prepared in ArcGIS environment with ISO-2Alpha and ISO 3-Alpha country codes. The primary data source for multiple socioeconomic indicators is the World Bank Data Centre [22], which manages global data for each country in the form of more than 2000 development indicators. After reviewing the literature, GDP per capita and Current Health Expenditure (% of GDP) were chosen as measures of Economy. For the strength of health infrastructure, hospital beds (per 1000 people), physicians (per 1000 people), and nurses (per 1000 people) were considered. As a representation of demography, population and the most vulnerable segment of the population, i.e., population ages 65 and above (A65abp) were selected. Data were also collected for disability-adjusted life year (DALY) from [23], which represents burden of diseases for every country. The urban population density for each country was also measured. In this regard, boundaries of urban areas from ESRI and a population density grid of 1x1 km from NASA [24] were downloaded. The average population density for each urban area was summarised and calculated and then the average of all urban areas in each country was computed. Disease vulnerability composite indexes (such as those mentioned in [20]) were not pre-selected as they mainly relate to the spread of diseases and not the mortality risk. COVID-19 data as of 13 May 2020 was acquired from JHU [3], which included four variables (active cases and three growth rates: Confirmed, recovered, and mortality). It was realised that these growth-related variables would effectively map COVID-19 current risk in a temporal sense as well. The fifth dynamic factor recognised was the susceptible population, which was considered as the percentage of the total population who was not infected yet.

In order to capture governmental intervention strategies, the stringency index proposed as the Oxford COVID-19 Government Response Tracker (OxCGRT) [25] was utilised as the sixth dynamic factor. This is defined as “a policy stringency index (calculated) by combining 13 policy indicators, including school and workplace closures, travel bans, as well as fiscal policy measures”. To allow for the measures to take effect, and to give increased importance for measures taken earlier rather than later, the modelling considered a weighted average stringency level for 13 May 2020 based on a formula provided in literature [26,27]. A total of 153 countries were represented in the analysis.

### 2.2. Modelling Techniques

In order to avoid scale diversification, prior factor normalisation is a common technique [28]. Normalisation was applied in three steps in the presented modelling setup. First, each variable other than the stringency and susceptible cases was converted into per capita or a larger unit of the population, such as per one thousand or million people. In the second step, Box-Cox transformation [29] was applied in R [30] for reducing the skewness in the data. Later in the third step, in order to manage the direction of association of factors with the mortality risk, 10 bins of each transformed variables were created. The bins were labelled on a 1–10 scale based on the direction of the significance to the risk (where 1 represent the lowest risk and 10 shows the highest risk). For instance, the higher the value of a factor represents a higher risk, so the labelled values follow the same trend i.e., lowest value of factor = 1 (the lowest value on the scale) and highest value of factor = 10 (the highest value on the scale). Conversely, based on the reverse relationship, labelled values are in the opposite direction i.e., the lowest value of a factor = 10 (highest value on scale) and the highest value of a factor = 1 (lowest value on the scale) (e.g., hospital beds). When a socio-economic or demographic factor was missing, it was replaced with its real value through a manual search of appropriate sources. All of the collected and transformed data were organised in ArcGIS and Excel files for easy interoperability between analytical and visualisation software.

The model was executed as a three-step strategy. Firstly, in order to visualise a base mortality risk assessment (or pre-COVID mortality risk scenario) a multi-criteria Analytic Hierarchy Process (AHP) [31] was used to compute weights (relative importance) for the nine static indicators. The pair-wise comparison in AHP is a common technique to assess the significance of each indicator [32] with a tolerable degree of inconsistency in each pairwise comparison [33]. The first and second author independently evaluated the relative importance of the factors and the discrepancies were accordingly resolved. The relative importance of weights was handpicked in accordance with the analysis of research literature, which has established the impact of various indicators on COVID-19 mortality [26]. The computed weights are summarised in a table (see Table 1). The baseline scenario represented the health risk in general terms without focusing on the COVID-19 pandemic. It showed the strength of each nation based on their economy, health infrastructure, and demography. Secondly, a multivariate linear regression model was conducted where the dependent variable was a normalised COVID-19 mortality for a country as of 13 May 2020. The independent variables were the nine static socio-economic factors described earlier. Thirdly, the regression model was repeated as mentioned in the second step but this time with the on-top addition of the six dynamic factors associated with COVID-19, giving a total of 15 independent variables. The third scenario that included COVID-19 related data alongside stringency data and static variables provided a reflection of the current pandemic state of the world.

For the regression models, the regression predictors were then assessed for relative importance via assigning of weights using the relaimpo package [34]. Lastly, the weights obtained from the modelling were aggregated with their ranks in the form of a weighted sum (see Equation (Equation 1)):(1)$$Risk_{i}=\sum_{j=1}^{n}w_{j}a_{ij}$$
where, *w* = weight, *a* = rank value, *i* represents each country, and *j* represents each factor value of *i*th country.

## 3. Results

The main elements of the analysis are discussed using the results from the three steps in the modelling. Firstly, a spatial map was visualised to model the base risk on a global scale (see Figure 1), such that the base risk was computed using weights derived from the Analytic Hierarchy Process or Pairwise comparison. Countries such as Japan, Norway, Germany, Switzerland, Austria, Belgium, Denmark, Sweden, Netherlands, and Finland were deemed to be at the highest mortality risk due to their large ratio of elderly population, emanating from the higher weight to the A65abp indicator in the Analytic Hierarchy Process. Several African countries were deemed to be less of a mortality risk due to their lower life expectancy and consequently lower proportion of elderly in the population. These countries included Ethiopia, Yemen, Sudan, Senegal, etc.

The second stage of the analysis was a linear regression model using nine static variables as independent factors and a COVID-19 normalised mortality on 13 May 2020 as the dependent variable. The results are referred to in a table (see Table 2). $R^{2}$ was such that it could explain 69% variance in the entire dataset. The ratio of the elderly in the population (or A65abp) emerged as a significant predictor. The GDP of countries and number of hospital beds were nearing that significance. Consequently, these predictors were also assigned higher relative weights by the relampo package in R; 19% for A65abp and 22% for GDP. As a means to check multicollinearity, the variance inflation factor for all predictor variables was lower than nine.

For the third step in the analysis, the regression modelling was repeated with the addition of six dynamic variables associated with COVID-19, giving a total of 15 independent variables. Then, the model was able to explain up to 88% variance in the data. The dynamic variables tended to heavily dominate over the static socio-economic factors with three dynamic factors having significant predictive power. The ratio of the elderly was yet again a significant predictor towards COVID-19 mortality risk as was the number of nurses. Furthermore, the government-enforced stringency level did not emerge as a significant predictor in this model. As a means to check multicollinearity, the variance inflation factor for all predictor variables (except GDP) was lower than 7. A table (see Table 3) is presented that summarises the top 10 countries sorted on the basis of the current mortality risk and their predicted risk ranking (both pre-COVID and on 13 May 2020) and the latter using the modelling analysis comprising of both static and dynamic indicators. The table shows that at least for this subset of 10 countries, they are at a COVID-19 mortality risk level where they were anticipated to be consistent with their baseline risk assessment.

A spatial map illustrates the mortality risk of COVID-19 as predicted by the third step of the analysis (see Figure 2). A spatial map was also drawn based on the change from baseline in COVID-19 mortality risk as projected from the linear regression modelling technique, which used a conglomerate of both static and dynamic factors (see Figure 3), essentially a difference between Figure 2 and Figure 3). The map clearly indicates that most countries were at a level of expected risk or lower risk on 13 May 2020 compared with what was originally predicted in the base scenario (noting how most countries are coloured in shades of yellow, orange, or green, which refers to a reduction or equivalence in risk from what was expected). All materials related to the modelling such as R code, output and base data is provided in the form of a Appendix A.

## 4. Discussion

The results show that only 44/153 countries experienced a more than 20% increase in COVID-19 related mortality risk as at 13 May 2020 when compared against their pre-COVID mortality risk assessment. Therefore, in general it can be ascertained that most countries have already experienced the worst of the pandemic and are now passing through lower levels of mortality risk. Eventually, either health systems have caught up with medications established, immunity has increased, or lockdown measures and quarantining has taken full effect resulting in a stability or decrease in mortality rate [35]. Furthermore, interesting yet subtle differences were observed in COVID-19 mortality risk in Western Europe. The pre-COVID base mortality risk showed that countries such as Germany, Belgium, and Switzerland were at a higher risk. However, the change in the mortality risk spatial map shows that most of these countries are now in a state of decreased risk. Countries such as Norway, the United Kingdom, and Spain also witnessed a decrease in mortality risk but not as extensive as other Western European countries. The nuances of social, cultural, and economic patterns in mainland Europe have been discussed in prior work [19], where not only is the ratio of the elderly in the population emphasised but their social habits and family structure are also emphasised (that is, living alone or in a larger family, where both can lead to contagion). In summary, the results show that the ratio of the elderly appears to be a strong predictor for eventual COVID-19 mortality risk. However, this may be considered in light of household layout or “co-residence patterns” [27,36], where in countries with inter-generational cohabitants, indirect infections and deaths may occur.

The risk change spatial map indicated that Central Africa was deemed to be in a position of higher risk and this may simply be that the peak of the pandemic was yet to hit in this region and had passed in regions such as Western Europe, North America, and far-Eastern Asia [37]. If developing countries such as those in Africa do not learn from the fate of those countries who suffered just recently due to the pandemic of COVID-19, they would be in further risk due to a lack of resources and weak health infrastructure. Prior work indicates through simulations that even though African countries may diagnose fewer absolute cases, they may face a higher mortality risk in comparison to European countries [27]. It is also noted that the predictive power of the regression model increased when a suite of both static and dynamic factors were used. The lower predictive power of the static socio-economic factors (only A65abp emerged as significant) highlights the need for a customised and tailored composite index for a risk assessment related to COVID-19, particularly for mortality. The weighted stringency level on 13 May 2020 of the countries in the sample was also not a significant predictor and this may have been due to the lower variation in the range of stringency globally. Around mid May, most countries had entered or were already in a state of advanced lockdown.

### Limitations

Like any study of this nature, there are limitations to the analysis, which is susceptible to any abnormalities and uncertainties in the data. For example, testing rates and measurement protocols differ across each country. All COVID-19 data were acquired through JHU where micro level data of COVID-19 patients are not available for all countries. If available, indicators such as the number of ventilators available would also be very useful and significant. Similarly in the future, including cultural aspects such as the diversity in ethnicity or what vaccines [38] are prevalent in the country may reveal new association trends with the risk of COVID-19. Furthermore, as with any global overview, a micro-level discernment of trends is dearly missed as countries that are complex in their makeup are placed together, as if homogeneous in one regression model. This was evident, as in the subtle mortality risk differences across mainland Europe. It is important to also acknowledge certain susceptibilities of the modelling. The risk assessment is only seemingly applicable as of early May 2020 and cannot control for secondary peaks of COVID-19 cases, which would lead to a transition of risk. In addition, the risk modelling originated from the weighted strategy of nine static indicators. Although there has been an effort to control for biases and follow an informed process, the method may be considered as subjective. Lastly, the paper presents an assessment of risk that is based on levels and not probabilities and this can also be addressed in future work.

## 5. Conclusions

In this paper, a mortality risk-based evaluation of COVID-19 on a global scale using data as at 13 May 2020 is presented. Using a multi-weighted approach, a range of unique scenarios using a mixture of static and dynamic variables were incorporated. The main finding was that the ratio of the elderly in a population clearly emerged as a significant mortality risk predictor for COVID-19, however this must be considered in light of the residency makeup of individual countries. In addition, a conglomerate of static socio-economic factors and dynamic factors associated with COVID-19 growth and spread had higher predictive capability. The current stringency of government-imposed restrictions was also not observed to have an impact. In general, as on 13 May 2020, from a spatial perspective the current mortality risk projections of COVID-19 may be considered as lower or as expected for most countries around the world.

## Figures and Tables

**Figure 1 ijerph-17-05592-f001:**
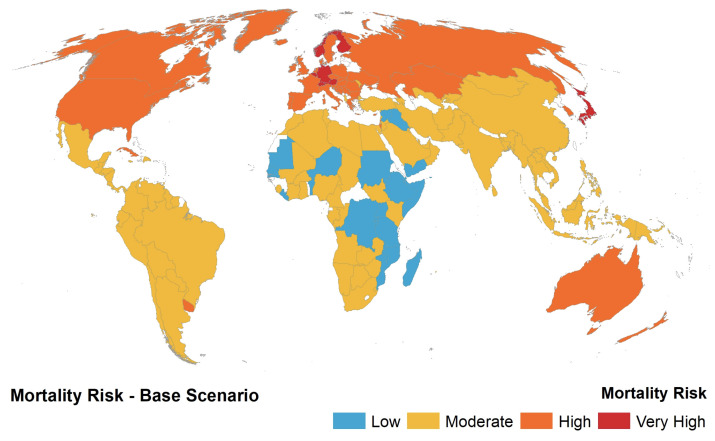
Base risk spatial map where each country had its risk normalised onto a range of 1–4, where 1 indicated as Low risk and 4 as Very High risk.

**Figure 2 ijerph-17-05592-f002:**
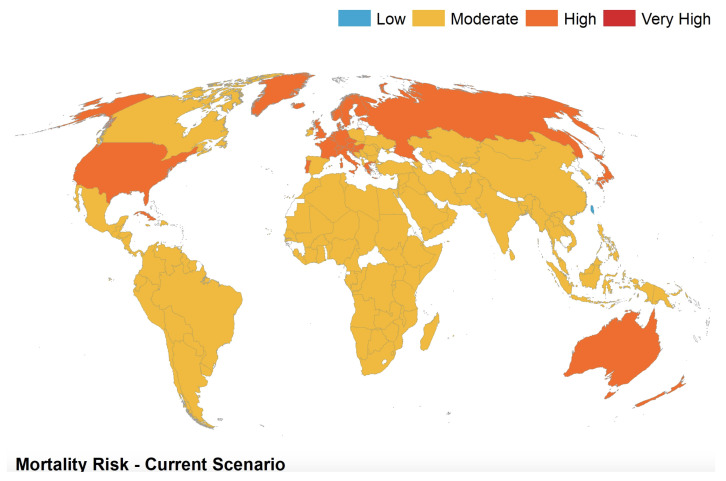
Current risk as indicated by both static and dynamic factors, where each country had its risk normalised onto a range of 1–3, where 1 indicates Low risk and 3 indicates High risk.

**Figure 3 ijerph-17-05592-f003:**
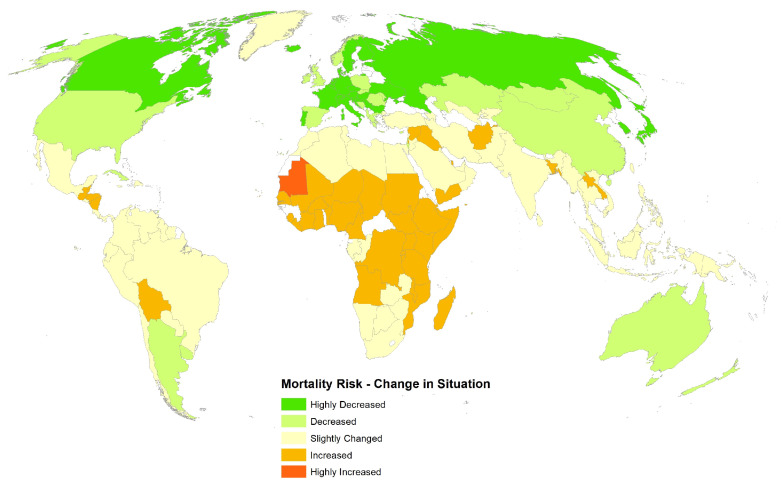
Change in risk where each country had its change of risk normalised onto a range of 1–5, where 1 indicates highly decreased risk and 5 indicates highly increased risk.

**Table 1 ijerph-17-05592-t001:** Base scenario weights of static factors.

Variable	Weight
Average Population Density	0.027
Population	0.039
Health Expenditure	0.058
GDP	0.09
DALY	0.157
Nurses	0.157
Physicians	0.157
Hospital Beds	0.157
A65abp	0.157
Consistency Ratio < 0.01	

**Table 2 ijerph-17-05592-t002:** Regression results for risk of mortality where for *p*-values “***” represents p<0.001 m, “**” represents p<0.01, and “*” represents p<0.05.

Regression Model	$R^{2}$	Significant *p*-Values	Top Weights
Static factors	0.69	A65abp ***	A65abp (0.19), GDP (0.22)
Static and dynamic factors	0.88	A65abp ***, nurses *, susceptible *, active ***, mortality growth **	active (0.20), susceptibles (0.15), mortality growth (0.11), A65abp (0.10)

**Table 3 ijerph-17-05592-t003:** Top 10 countries ranked on actual mortality rate and their predicted risk assessment.

Country Name	Mortality Rate (Actual)	Pre-COVID-19 Mortality Risk Rank (Predicted)	COVID-19 Mortality Risk Rank as at 13 May 2020 (Predicted)
San Marino	1213.6	41	3
Belgium	774.2	7	8
Andorra	636.3	46	60
Spain	580.1	35	41
Italy	514.1	14	17
United Kingdom	499.1	25	16
France	403.5	11	13
Sweden	339.8	9	11
Netherlands	322.8	10	12
Ireland	308.4	27	33

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
