# Peer review of "COVID-19 Global Risk: Expectation vs. Reality"

_ijerph, 2020, doi:10.3390/ijerph17155592_

Round 1

Reviewer 1 Report

Thanks for the invite to review this manuscript!

This is a good work. But it is not ready for my recommendation of its publication at IJERPH. There are several reasons. I start with the abstract. The abstract is not adequate. It must describe, in brief, all major sections: backgrounds, methods, results/findings, and finally conclusions. 

The manuscript seems to describe the data collection, the methods, and the results and discussion reasonably well. However, I am a little bit concerned as to why the authors did not provide with the data (that they have compiled from various sources as then mention in the paper) for 154 countries in Supplementary Material, given that journal they aim to publish the work is an open access journal. Also, it is not clear how the weights given in Table 1 were derived. (Note, in the main text the authors say, see Table x: they need to do good proofreading of the manuscript as there are typing mistakes here and there.) To a certain extent, the authors have tried to explain it, but a clear description of all the steps involved in it is required for transparency and reproducibility of the results by peer groups.

It is not clear as to why all 9 static factors have the same weights across all 154 countries. Or did I miss something? The authors need to make it clear. A minor thing. In the main text, in the last paragraph under the Introduction section, the authors use a phrase "paramedic staff," but this phrase does not make it to Table 1. Why? In addition, why do the last 5 static factors (Table 1) have the same weight (of 0.157)? This is not discussed anywhere, even not as one of the limitations. This is the reason (in my view) that the authors may need to provide a clear description of how the weights were calculated.

In my opinion, the results could be described in a little more elaborate way. For example, the authors say, "the model was able to explain up to 88% variance in the data." I do know that the authors aimed to perform the analyses at the global scale. But could not they have done some (select) country-level analyses. How will this explaining of "up to 88%" hold for the USA, Italy, the UK, the China, to name a few select countries?

When I looked at Table 2, I found that there are some additional discrepancies in Table 1. In Table 2, A65abp has weight of 0.32. A65abp does not appear in Table 1. Why? Also, why GDP has its weight at 0.27. Table 2 needs to be explained clearly.

The authors mention a policy stringency index. Did they calculate this index for all 154 countries? If yes, then where are the indices? I strongly suggest that the indices should be a part of Supplementary Material.

The authors say, "We replaced all missing values with 0 for nullifying the impact on final aggregation." How this replacement with 0 would impose some kind of limitation for generalization of their findings. This may need some discussion as a part of the limitations.

The following sentence reads: "We organized all collected and transformed data in ArcGIS and Excel files for easy interoperability between analytical and visualization software." Have the authors made these files available to the journal?

Reviewer 2 Report

In the growing literature on COVID-19-related studies, this paper takes a more global approach in estimating expected risks of mortality. However, there are several issues that need to be resolved before the paper is of good-enough quality to be published in a high-impact journal such as IJERPH.

The text contains inaccuracies that need to be corrected. E.g.

Lines 23-24. The demographic attributes of any country such as it’s socio-economic factors ultimately dictate not only how a pandemic may spread ...
Demographic attributes (age and sex structure, for instance) are one thing, socioeconomic factors (education, income, ...) are another. Social scientists sometimes do talk about sociodemographic factors.

Lines 35-36. Although the highly cited article by Jones et al uses population density as a socio-economic factor in their study, they also mention in their article that population density is a socioeconomic "driver", which I think is a more accurate description. There are many examples of very poor but also very densely populated countries in the world (e.g. the Indian subcontinent), and conversely, of very wealthy but sparsely populated countries (e.g. Nordic countries, New Zealand, Australia, Canada). On the other hand, within countries, larger urban areas could drive economic growth. The authors should therefore better explain the relevance of population density as a socioeconomic factor in the context of the spreading of SARS-CoV-2 (and not necessarily COVID-19).

Lines 54-57. Prior work has attempted to understand the correlation between certain risk elements such as demographic variables and certain specific socio-economic factors (such as GDP and population density) towards COVID-19 but this was only restricted to China.
This is not true. A pre-print article by Mogi and Spijker published in April (The influence of social and economic ties to the spread of COVID-19 in Europe; https://doi.org/10.31235/osf.io/sb8xn) is one example (they also cite other studies) and by now (early July) there are others as well.

Please provide an appendix with the values of the observed vs expected number of COVID-19 deaths based on the modeled predictions.

Please make the "discussion" a separate section (Section 4) from the results (Section 3).

I notice from the results that several of the European countries that show the highest mortality risk (Germany, Austria, Norway, Finland) are actually the countries where mortality was quite low when compared to other European countries, especially those in Southern Europe (Italy, Spain) where multigenerational households are much more common and population ageing is at a more advanced stage. These, and other anomalous results, should be discussed, especially in the context of the title of the article: COVID-19 Global Risk: Expectation vs Reality.

Please also discuss your results in the context of the findings recently published by other reseachers. For instance, Esteve et al (2020) studied from harmonized census data from 81 countries how age and coresidence patterns shape the vulnerability of countries’ populations to outbreaks of coronavirus disease 2019 (COVID-19). Specifically, they estimate variation in deaths arising due to a simulated random infection of 10% of the population living in private households and subsequent within-household transmission of the virus and found that preventing primary infections among the elderly is the most effective in countries with small households and little intergenerational coresidence, such as France, whereas confining younger age groups can have a greater impact in countries with large and intergenerational households, such as Bangladesh. In this article, the authors will also find other useful papers they can refer to (also for their introduction section).
Esteve et al (2020) "National age and coresidence patterns shape COVID-19 vulnerability". Proceedings of the National Academy of Sciences (PNAS) Jun 2020, 202008764; DOI: 10.1073/pnas.2008764117

Line 129 Please state which table is referred to (Table x)

Lastly, the text contains numerous awkward sentences that need to be tweaked by a native speaker. Just to show three (see especially the parts in bold):

Lines 33-35 In prior literature, a number of interesting case studies are presented analysing the numerous socio-economic risk factors that dictate the epidemiology of various pandemics or epidemics other than COVID-19.

Lines 52-54 - our aim is to assess the mortality risk level of each country to a proposed epidemic (i.e. COVID-19) as determined by a collage of objective yet static socio-economic factors, in combination with a range of dynamic epidemiological indicators.

Lines 113-114 Later in third step for managing the direction of association of factors with the mortality risk we created ten bins of each transformed variables.

Round 2

Reviewer 1 Report

There are still some improvements/clarifications required throughout the manuscript.

Abstract: Under the Conclusion section, the authors say, "We conclude with the main limitations of our analysis and suggest future avenues of data acquisition related to COVID-19."

In this section, the authors must write out, in brief, what is/are their main conclusion(s), based on their findings, along with major limitation(s) if any. This one sentence the way it is written is not the major conclusion or limitation of the paper. (Note the conclusion section (of an abstract) is for the conclusion(s), along with major limitation(s) if any. The section is definitely not for authors to just say that we conclude by our conclusions/limitations.)

I have a minor comment: the authors should pay a little more attention to their use of punctuations. One example. In the sentence at lines 26-27, the phrase "named severe acute respiratory syndrome coronavirus-2 (SARS-CoV-2)" should end with a comma (,). This type of mistakes can be caught if proofreading is done properly.

An example of not doing proofreading carefully. "All COVID-19 data was acquired through JHU where micro level data of COVID-19 patients is not available for all countries." The auxiliary verb "was" and "is" should have been replaced with, respectively, were and are.

The figure captions of Figures 1, 2, and 3 are not adequate. What do "low," "moderate," "high," and "very high" mean? Can the authors not describe then in the captions of Figures 1 and 2? Ditto for the figure 3 color legends.

I am happy to see that the authors have provided a supplementary file which has further details on the methods section. I congratulate the authors for doing this. I request them to take a careful look at the various sections in this file to make sure that there is no mismatch between what is provided in this supplementary information file and the methods described in the main text.

Author Response

We are once again very appreciative of the reviewer's comments and feedback in the latest round. As per their recommendation the following edits were carried out:

  • The abstract was amended, in particular the conclusion sub-section within the abstract.
  • Cross check of supplementary file against data showcased in manuscript
  • Addition of required details in the Figure captions
  • Fixing of mentioned typos and editorial changes in the review
  • On request of the reviewer, we have asked a native English speaking colleague to proof read our manuscript. This has led to many changes of editorial nature, in particular complete removal of all first person references (we, our, us).

Reviewer 2 Report

The new version is an improvement of the first one, but I still have a couple of comments:

lines 31-32. To "The demographic attributes of any country such as population ..." please add "size and density" and add after "age" "structure".

References: To the Mogi and Spijker paper, please add "SocArxiv Prepint at https://osf.io/preprints/socarxiv/sb8xn/" after 2020.

Lastly, the text still contains numerous awkward sentences that need to be tweaked by a native speaker. I compared the documents carefully but did not see much (if any) improvement in writing style. I therefore doubt that the article has been properly proof-read. I give the authors one recurring error in the text that a native speaker would never make: confuse the word "it's" with "its". The former is a short form of "it is" or "it has", the latter is a possessive pronoun (http://englishplus.com/grammar/00000227.htm). See for instance lines 32 and 34. In fact, even if in line 32 you would have used "its", the sentence still wouldn't make sense (the "socio-economic factors" of whom?).
So, there are still many awkward sentences. For instance, another one that makes no sense to me is the sentence that starts with "Considering a (lines 35-38).
Please find someone to go through the text carefully. If it will take more time than the stipulated deadline, just ask the editors for an extension.

Author Response

We are once again very appreciative of the reviewer's comments and feedback in the latest round. As per their recommendation the following edits were carried out:

  • Fixing of mentioned typos and editorial changes in the review (including; grammatical errors in lines 31-38 and Mogi reference details)
  • On request of the reviewer, we have asked a native English speaking colleague to proof read our manuscript. This has led to many changes of editorial nature, in particular complete removal of all first person references (we, our, us).